ecology, evolution

evolutionary transitions, fitness trade-offs, life history, macroevolution, tetrapod

**Author for correspondence:**
Cecina Babich Morrow
e-mail: babichmorrowc@gmail.com

# Macroevolution of dimensionless life-history metrics in tetrapods

Cecina Babich Morrow[1,2,3], S. K. Morgan Ernest[4] and Andrew J. Kerkhoff[3]

[1]Spring Health, New York, NY, USA
[2]Center for Biodiversity and Conservation, American Museum of Natural History, New York, NY, USA
[3]Department of Biology, Kenyon College, Gambier, OH, USA
[4]Department of Wildlife Ecology and Conservation, University of Florida, Gainesville, FL, USA

CB, 0000-0003-2188-1495

Life-history traits represent organisms' strategies to navigate the fitness trade-offs between survival and reproduction. Eric Charnov developed three dimensionless metrics to quantify fundamental life-history trade-offs. Lifetime reproductive effort (LRE), relative reproductive lifespan (RRL) and relative offspring size (ROS), together with body mass can be used to classify life-history strategies across the four major classes of tetrapods: amphibians, reptiles, mammals and birds. First, we investigate how the metrics have evolved in concert with body mass within tetrapod lineages. In most cases, we find evidence for correlated evolution among body mass and the three dimensionless metrics. Second, we compare life-history strategies across the four classes of tetrapods and find that LRE, RRL and ROS delineate a space in which the major tetrapod classes occupy mostly unique subspaces. These distinct combinations of life-history strategies provide us with a framework to understand the impact of major evolutionary transitions in energetics, physiology and ecology.

## 1. Introduction

Life-history traits quantify the two crucial components of fitness: survival and reproduction. A species's life-history strategy (i.e. how it allocates resources to survival and reproduction) impacts its fitness, and thus its success in terms of population growth and extinction risk [1–3]. Because resources are limited, organisms cannot optimize both individual survival and reproductive investment, so allocation to one component of life-history necessitates trade-offs in other areas [4,5]. Organisms navigate these constraints in a variety of ways to maximize overall fitness, creating an astonishing diversity of life-history strategies. Life-history theory is an attempt to understand how natural selection generates this diversity based on fundamental trade-offs in survival and reproduction [4]. Life-history parameters typically have units of mass or time, making it difficult to compare the strategies of small, short-lived organisms to large, long-lived ones. Comparing life histories across variations in sizes and paces of life requires an approach that distills strategies into comparable yet biological meaningful metrics.

In a series of publications [5–8], Eric Charnov proposed classifying and comparing life histories based on three particular dimensionless metrics that represent the fundamental trade-offs organisms must navigate in order to maximize fitness [8]: (i) lifetime reproductive effort (LRE), which measures the proportion of adult mass that a female will allocate to offspring over her lifespan [8]; (ii) relative reproductive lifespan (RRL), which quantifies time to maturity relative to the total amount of time available for reproduction [7,8]; and (iii) relative offspring size (ROS), which is the ratio of offspring to adult size [9].

Charnov originally proposed that some (but not all) of these dimensionless metrics might be invariant with body mass for some (but not all) taxa [8], but a

subsequent debate about invariance [10,11] has obscured the fact that, regardless of whether they are all truly 'invariant' [12], these dimensionless metrics capture fundamental life-history trade-offs [8]. LRE represents the energetic trade-off of reproductive effort versus adult mortality, RRL captures the trade-off in time spent in growth/development versus reproduction and ROS is related to trade-offs in the size, number and survivorship of offspring. Moreover, removing the magnitudes of mass and time allows for the comparison of life histories on a common scale across groups of organisms with large variation in body size. Charnov hypothesized that variation in these metrics would be greater between major groups of organisms than within groups, since organisms within a group share similar life-history constraints [8].

Here, we use Charnov's dimensionless life-history metrics to explore how life-history strategies vary both within and across clades of tetrapods. To conduct comparative analyses between groups of organisms, Charnov envisioned a 'life-history cube' defined by LRE, RRL and ROS as axes, with different groups of organisms occupying different regions of this trait space [8]. While we are not primarily concerned with whether the dimensionless metrics are invariants within taxa, we retain mass as a fourth axis since adult body mass is known to strongly impact many life-history traits, including components of the dimensionless metrics. If Charnov's hypothesis is correct, different tetrapod lineages should occupy distinct regions of the resulting 'hypercube', despite substantial variation in body size within and between lineages.

Tetrapods are an ideal group for this study because they are distinguished by three major adaptations—the amniotic egg, endothermy and flight—that strongly impact adult survival and allocation to reproduction. Amniotes produce larger eggs with higher rates of respiration [13] and more substantial yolks [14], yielding more developed offspring and potentially increasing juvenile survival rates and reproductive allocation. Endotherms can attain greater metabolic power and rates of production [15] and exploit a wider range of environments [16], but they also face energetic costs that limit their minimum size. Without these constraints, ectotherms can potentially display a wider variety of life-history strategies in response to their local environmental conditions. Finally, flight reduces predation risk, decreasing extrinsic mortality rates and lengthening the lifespans of volant organisms [17,18]. But flight also imposes higher parental investment costs [19] and is energetically costly itself, which can potentially limit reproduction. By altering constraints on investment in reproduction and survival, these key adaptations may have impacted the evolution of life-history strategies. We consider how they may influence the position of amphibians, reptiles, mammals and birds in life-history trait space.

## 2. Methods

### (a) Data
We compiled life-history trait data for birds, mammals, reptiles and amphibians from multiple sources to calculate the dimensionless life-history metrics. For the birds and mammals, we used data exclusively from the Amniote Life History Database [20]. For the reptiles, we supplemented the data available in Amniote with another published set of reptile life-history traits [3] through a two-step process. First, if a reptile species present in the Amniote database lacked trait data for one of the life-

history traits necessary to calculate the dimensionless metrics, we filled in the corresponding value from Allen et al. [3]. Second, we added trait data for species present in the Allen et al. database but not in Amniote. For the amphibians, we obtained life-history trait data from the AmphiBIO database [21].

### (b) Calculation of dimensionless metrics
We used the combined amniote and amphibian data to calculate the three dimensionless life-history metrics for 1650 tetrapod species, including 171 birds, 842 mammals, 491 reptiles and 113 amphibians. For further detail on the precise calculations for each metric, see the electronic supplementary material.

### (i) Lifetime reproductive effort
The first dimensionless metric, LRE, is the product of reproductive effort and average adult lifespan, where reproductive effort is average reproductive allocation per unit time ($R$) divided by average adult body mass [8]. To calculate $R$, we multiplied litter or clutch size by the number of litters or clutches per year and then multiplied this value by the mass of offspring at independence. We divided $R$ by adult body mass to calculate reproductive effort and multiplied the result by adult lifespan (the time from reproductive maturity to death) to calculate LRE:

$$\text{LRE} = \frac{\begin{array}{c}\text{litter size } (n) \times \text{litters per year } (n/\text{yr}) \times \\ \text{mass at independence } (g)\end{array}}{\text{adult body mass } (g)}$$
$$\times \text{ adult lifespan (yr)}$$

To determine mass at independence, we used mass at fledging for birds, weaning for mammals, hatching for reptiles and offspring or egg for amphibians (whichever value for offspring mass was provided in AmphiBIO). While Charnov's model calls for an average body mass, AmphiBIO only provides a maximum adult body mass, so we used this value to provide an approximate value of this metric for amphibians [21]. In order to ensure that this decision did not bias the analyses, we also performed amphibian analyses by converting the minimum and maximum lengths from AmphiBIO to body mass using allometric equations for Anura and Caudata [22]. We used the models predicting mass from SVL for Anura and Caudata, rather than those including habitat and paedomorphy since the majority of frog species in AmphiBIO existed in multiple habitats and paedomorphy was not reported for the salamander species. We then averaged the calculated minimum and maximum body masses to find an average adult body mass for amphibians. Since results did not differ based on the value of body mass used (electronic supplementary material, figures S1 and S2, table S1), we present results using the maximum body mass in the body of the paper and include the results of the average mass converted from SVL in the electronic supplementary material. We used maximum longevity, rather than average longevity, to calculate adult lifespan for all classes due to data quality and availability.

### (ii) Relative reproductive lifespan
To calculate RRL, we divided adult lifespan by the time to female maturity:

$$\text{RRL} = \frac{\text{adult lifespan (yr)}}{\text{time to sexual maturity (yr)}}.$$

### (iii) Relative offspring size
In order to calculate the final dimensionless metric, ROS, we divided mass at independence by average adult body mass:

$$\text{ROS} = \frac{\text{mass at independence } (g)}{\text{adult body mass } (g)}.$$

We used the same criteria for independence for birds, mammals, reptiles and amphibians as used to calculate $R$.

## (c) Phylogenetic trees

To investigate the evolution of the life-history metrics between clades, we used a variety of published tetrapod phylogenies. For the mammals, we used the Fritz et al. [23] species-level supertree with the best date estimates. For the birds, we used the dated phylogeny of extant bird species published by Jetz et al. [24], constructed using the Hackett et al. [25] backbone [24]. Since reptiles are a paraphyletic group, we restricted our phylogenetic analyses to Squamata, the most diverse reptile order, using a time-calibrated phylogeny [26]. Finally, for the amphibians, we used a congruified timetree from the PhyloOrchard package [27], using Alfaro et al.'s timetree of gnathostomes [28] as the reference and the Pyron and Wiens amphibian phylogeny as the target [29].

In order to visualize evolutionary changes in life history across the four clades, we stitched together these four phylogenies to create a tetrapod phylogeny. We used divergence estimates from the TimeTree of Life [30] to combine the individual phylogenies according to a pipeline used by Uyeda et al. [31].

## (d) Analyses of life-history evolution within clades

Charnov proposed that, depending on the allometric relationships of the underlying dimensional parameters, some of the dimensionless metrics could be invariant with adult body mass for some taxa [32]. To test this, we performed phylogenetic least-squares regression analysis (PGLS) [33] to examine the relationship between each of the three dimensionless metrics and body mass. We conducted PGLS using the branch-length transformation indicated by the best-fit model of body mass evolution for each class. We fit Brownian motion, Ornstein–Uhlenbeck, Pagel's $\lambda$ and kappa models of natural log body mass using the GEIGER package in R [34]. To determine the best-fit model, we ranked by Akaike information criterion (AIC) and selected the model with the lowest AIC value. PGLS was implemented using the nlme and ape packages in R [35,36].

Because selection acts on integrated organism phenotypes, it is likely that components of life history do not evolve independently. To test for patterns of correlated evolution within tetrapod clades, we used multivariate phylogenetic linear models of LRE, RRL and ROS as multivariate responses with body mass as the predictor in each of the four tetrapod clades. We used the mvMORPH package to fit generalized least-squares models, using Pagel's $\lambda$ as the evolutionary model [37]. The Pagel's $\lambda$ evolutionary model allows the resulting phylogenetic models to accommodate a range of phylogenetic signal compared to Brownian motion models and permits the simultaneous estimation of the regression model and phylogenetic signal [38] based on Pagel's $\lambda$ [39]. The mvMORPH implementation of multivariate PGLS also provides the option to estimate measurement error as a nuisance parameter during the fitting process; we used this option as recommended in all applications with empirical data [40,41]. Because shared correlations with body mass could cause 'spurious' correlations among the life-history metrics, we used adult body mass as a predictor variable and calculated the evolutionary correlations between each of the three life-history metrics after controlling for body mass, based on the precision matrices of the models.

## (e) Comparisons of life-history between clades

We used multiple approaches to test Charnov's fundamental prediction that different taxa would occupy distinct regions of the life-history space described by the dimensionless metrics [8]. First, treating each metric separately, we used ANOVA to ask whether life-history variation within tetrapod clades was small compared to variation between clades. Because the deep relationships among the tetrapod clades are well established, we treated each class as phylogenetically independent and compared the four groups using standard ANOVA.

Second, we qualitatively examined macroevolutionary changes in the life-history metrics among tetrapod classes by simulating trait evolution along the branches of the tetrapod supertree. Trait values at each node were estimated based on a Brownian motion model, implemented using the phytools package in R [42]. These simulations provide a visual, qualitative depiction of trait variation within clades as well as shifts in trait values between clades of tetrapods.

Finally, in order to quantify life-history variation and overlap across clades, we used hypervolume analyses [43,44]. We created four-dimensional hypervolumes for the four major classes of tetrapod (birds, mammals, reptiles and amphibians) with adult body mass and the three dimensionless metrics as axes. All four axes were natural log-transformed for analysis. All hypervolumes were created using the hypervolume R package using the Gaussian KDE method with the default Silverman bandwidth estimator [44]. To compare hypervolumes between groups, we quantified individual clade hypervolumes and pairwise overlap among clades, using Sorensen similarity (as calculated by twice the volume of the intersection between the hypervolumes divided by the sum of the volumes of each hypervolume [45]) and the fraction of unique volume. The size of these hypervolumes represents the total diversity of life-history strategies for a given class, while the overlap indicates the similarity in strategies between classes.

# 3. Results

## (a) Evolution of life histories within clades

There is strong evidence of phylogenetic structure for most of the life-history metrics within most of the lineages, based on our calculations of Pagel's $\lambda$ (table 1). Body mass showed a strong phylogenetic signal, with Pagel's $\lambda$ values close to 1 in all four groups of tetrapods. In amphibians, neither LRE nor RRL demonstrated any phylogenetic signal, although ROS did. For all other groups of tetrapods, all three dimensionless metrics showed a phylogenetic signal. Mammals generally exhibited the strongest phylogenetic signal across each of the life-history metrics, followed by birds and then squamates.

The dimensionless life-history metrics also exhibited correlated evolution with body mass across the different clades. Ectotherm body size evolution was best-fit by the Ornstein–Uhlenbeck model, while endotherm body size evolution was the best-fit by the Pagel's $\lambda$ model. After accounting for evolutionary relationships, body mass was negatively correlated with LRE in amphibians and mammals ($p < 0.05$), but there was no significant relationship between the squamates and birds ($p > 0.05$) (table 2). RRL, on the other hand, was only significantly correlated with body mass in the squamates ($p = 0.0155$; table 2). Body mass was negatively correlated with ROS across all the clades after accounting for phylogeny ($p < 0.05$). The magnitude of the slope of this relationship decreased from the oldest class (Amphibia) to the youngest (Aves) (table 2).

The coefficients of the multivariate models were very close to those estimated by the univariate PGLS models above (electronic supplementary material, table S3). Based on the evolutionary covariances of these models, the dimensionless metrics exhibit some correlations even after controlling for the effects of adult body mass (table 3). LRE

**Table 1.** Pagel's $\lambda$ values with their associated $p$-values for each of the three dimensionless life-history metrics within amphibians, squamates, mammals and birds.

|  | body mass | LRE | RRL | ROS |
|---|---|---|---|---|
| Amphibia | 0.964 ($p = 0.001$) | 0.369 ($p = 0.56$) | $6.65 \times 10^{-5}$ ($p = 1.00$) | 0.873 ($p = 0.001$) |
| Squamata | 0.977 ($p = 0.001$) | 0.176 ($p = 0.001$) | 0.328 ($p = 0.001$) | 0.619 ($p = 0.001$) |
| Mammalia | 0.997 ($p = 0.001$) | 0.957 ($p = 0.001$) | 0.989 ($p = 0.001$) | 0.902 ($p = 0.001$) |
| Aves | 1.00 ($p = 0.001$) | 0.897 ($p = 0.001$) | 0.817 ($p = 0.001$) | 0.997 ($p = 0.001$) |

**Table 2.** Coefficients for PGLS using natural log body mass to predict each natural log-transformed dimensionless metric. PGLS was performed using a correlation matrix based on the model of body mass evolution with the lowest AIC for each class: Ornstein–Uhlenbeck for amphibians and squamates and Pagel's $\lambda$ for mammals and birds. For the full table of coefficients and model parameters, see electronic supplementary material, table S2.

| metric | effect of log body mass (s.e.) |
|---|---|
| LRE | |
| Amphibia | −0.29 (0.11)** |
| Squamata | −0.028 (0.032) |
| Mammalia | −0.17 (0.024)**** |
| Aves | 0.056 (0.065) |
| RRL | |
| Amphibia | 0.12 (0.062) |
| Squamata | 0.074 (0.030)* |
| Mammalia | 0.0093 (0.022) |
| Aves | 0.048 (0.048) |
| ROS | |
| Amphibia | −0.80 (0.11)**** |
| Squamata | −0.38 (0.019)**** |
| Mammalia | −0.16 (0.013)**** |
| Aves | −0.11 (0.027)**** |

*$p < 0.05$; **$p < 0.01$; ***$p < 0.001$; ****$p < 0.0001$.

and RRL exhibited moderately strong positive correlations within each class. LRE was also positively correlated with ROS in all classes, although the correlation between the two metrics in birds was relatively weak. In all classes, RRL and ROS were uncorrelated.

## (b) Comparing life histories across clades

As measured by the three dimensionless metrics, life-history strategies differed substantially across the four tetrapod classes. LRE increased from amphibians to reptiles to mammals, and finally to birds (ANOVA; $F = 1017.8$; d.f. = 3; $p < 2.2 \times 10^{-16}$; figure 1b). ROS demonstrated a similar pattern to LRE: mean natural log ROS differed across all four classes (Tukey HSD; $p < 0.05$), increasing by 98.1% from amphibians to birds (figure 1d). RRL also differed between all four classes (ANOVA; $F = 242.9$; d.f. = 3; $p < 2.2 \times 10^{-16}$; figure 1c). Mammals had the highest mean log RRL value, which was 16.7% higher than that of birds, 88% higher than that of reptiles and 114% higher than that of amphibians.

When viewed on the tetrapod phylogeny, LRE shows distinct shifts in values across the four classes: higher values of LRE appear to have evolved both in birds and mammals, while the amphibians display consistently low values (figure 2a). In comparison, RRL does not vary as much between classes, but, in general, the lowest values are found in amphibians and reptiles (figure 2b). There are certain clades of mammals, however, like the family Soricidae, which have RRL values comparable to or lower than those found in amphibians and squamates. In the birds, as well, certain Charadriiformes have quite low RRL values. Of the three metrics, ROS shows the most dramatic shifts across the four classes of tetrapods (figure 2c), with the lowest values in amphibians, followed by squamates and finally mammals and birds.

Even though the four tetrapod classes overlap in values for several of the individual life-history metrics, as well as body mass, they occupy relatively distinct regions of the space delineated by the dimensionless metrics (figure 3), as predicted by Charnov [8]. The trait space hypervolume, representing diversity in trait combinations present for a given group, decreased dramatically in size across classes in order of evolution. The two ectothermic hypervolumes, Amphibia and Reptilia, were 4.81 and 3.06 times larger, respectively, than the mammal hypervolume, while the bird hypervolume was 17% of the size of the mammal hypervolume. The class hypervolumes also occupied highly distinct regions of the trait space. The birds and mammals had the most overlap, with a Sorensen similarity of only 0.14 (table 4). The amphibian hypervolume was the most unique, not overlapping with the endotherm hypervolumes at all, and only having a Sorensen similarity of 0.027 with the reptiles (table 4).

We also compared the position of Chiroptera in trait space relative to the mammals and birds to examine the potential effects of flight on these life-history metrics (electronic supplementary material, figures S3 and S4). Chiroptera displayed very similar ranges of body mass, LRE and ROS to birds, although they showed higher RRL values (electronic supplementary material, figures S3 and S4; figure 4). The mammal and bat hypervolumes had the greatest overlap, with a Sorensen similarity of 0.28, while the birds and bats had a similarity of 0.19 (electronic supplementary material, table S4). The bats and the birds thus had higher similarity to each other than any of the total class hypervolumes had to each other (table 4).

## 4. Discussion

Charnov's dimensionless life-history metrics (LRE, RRL and ROS) provide a framework to compare organisms' life-history strategies across a range of body masses. The three

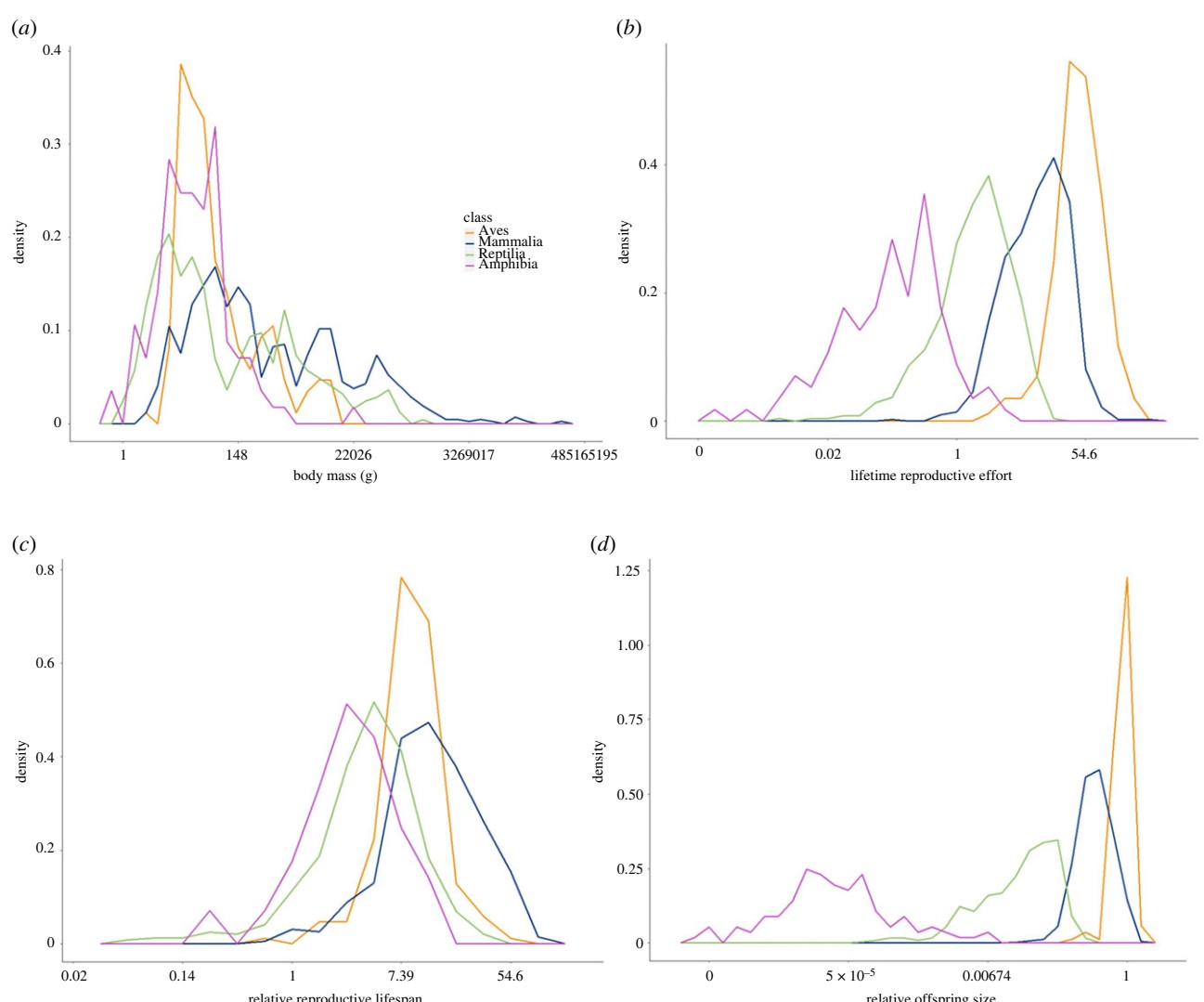

**Figure 1.** Frequency polygons of (*a*) natural log-transformed body mass (g), (*b*) LRE, (*c*) RRL and (*d*) ROS for tetrapod species with values for all three of the dimensionless traits. (Online version in colour.)

**Table 3.** Evolutionary correlation coefficients for each of the three dimensionless metrics after controlling for relationship with body mass. These coefficients were calculated from evolutionary covariance matrix of multivariate phylogenetic generalized least-squares models.

| Amphibia | RRL | ROS |
|---|---|---|
| LRE | 0.500 | 0.459 |
| RRL | — | −0.0268 |
| **Squamata** | **RRL** | **ROS** |
| LRE | 0.692 | 0.516 |
| RRL | — | 0.140 |
| **Mammalia** | **RRL** | **ROS** |
| LRE | 0.600 | 0.584 |
| RRL | — | −0.00365 |
| **Aves** | **RRL** | **ROS** |
| LRE | 0.726 | 0.222 |
| RRL | — | −0.0626 |

dimensionless metrics show a range of patterns of correlated evolution, which drive their relationships with mass in extant species. Furthermore, the major tetrapod classes display

unique combinations of these metrics (figure 3). The differences in subspaces occupied by each class may reflect the effects of crucial evolutionary transitions in energetics, physiology and ecology. There are by necessity only a few evolutionary transitions between tetrapod classes to examine, so we cannot establish any definitive link between evolutionary transitions and corresponding shifts in life-history strategy. Critically examining how the metrics vary between clades, however, can shed light on whether these metrics respond to evolutionary trade-offs in the way we would expect given Charnov's framework. By observing how the metrics change in tandem with major adaptations, such as the evolution of the amniotic egg, endothermy and flight, we explore the possible ecological and evolutionary influences on life history and the ways in which LRE, RRL and ROS can be used to characterize shifts in life-history strategy.

## (a) Amphibians to amniotes: evolution of the amnion

Amphibians are notably different from the amniote groups (reptiles, birds and mammals) in their life histories. Amphibians take longer to reach reproductive maturity relative to their total reproductive lifespan (lowest mean RRL; figure 1*c*), invest much less in reproduction (lowest mean LRE; figure 1*b*) and produce smaller offspring at independence relative to

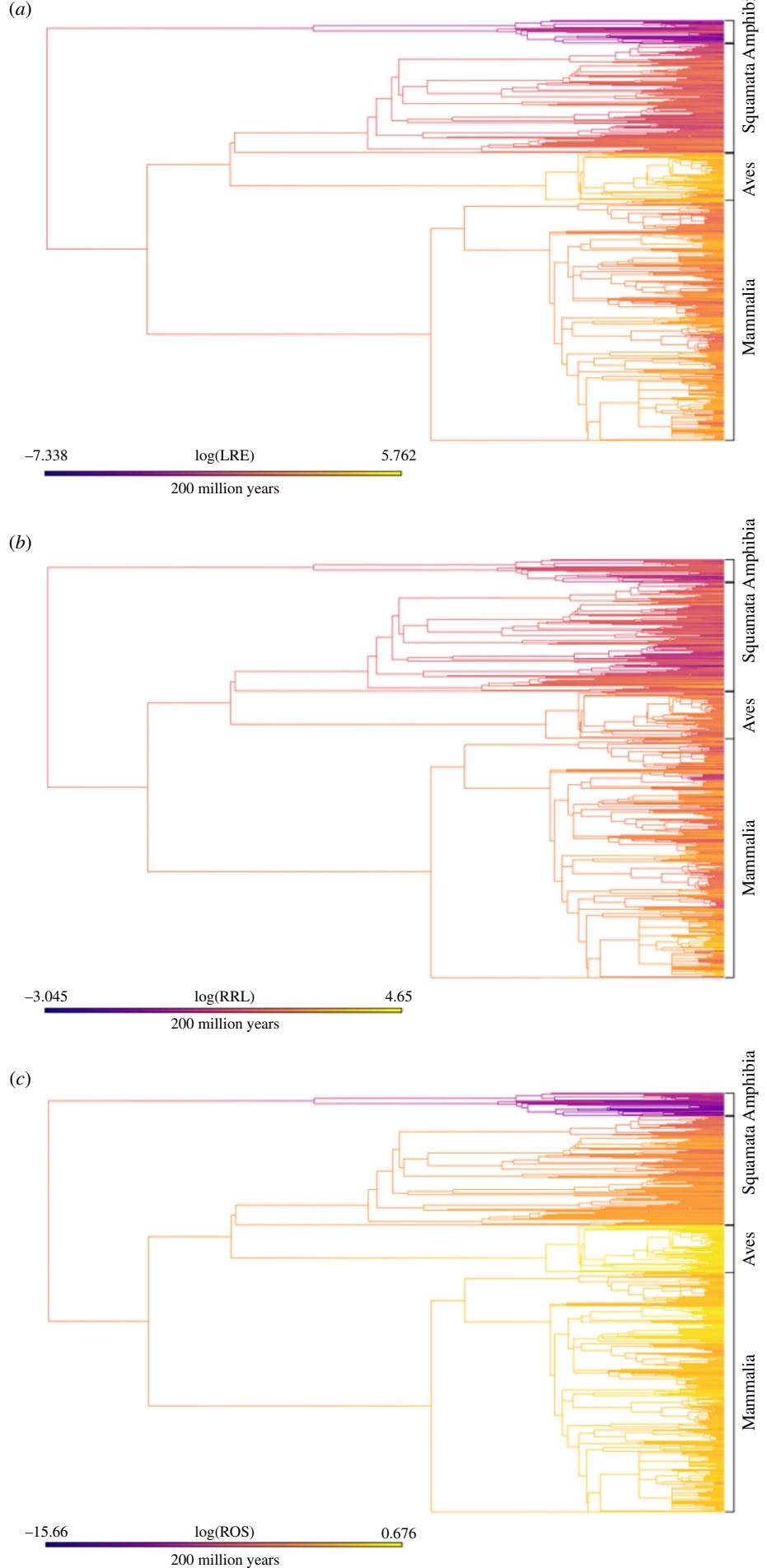

**Figure 2.** Plots of (*a*) LRE, (*b*) RRL and (*c*) ROS across a supertree of tetrapods. Trait values along the edges and at nodes were estimated based on a Brownian motion model of evolution. The colour ramp bar serves as a legend for trait values and a scale for branch lengths. (Online version in colour.)

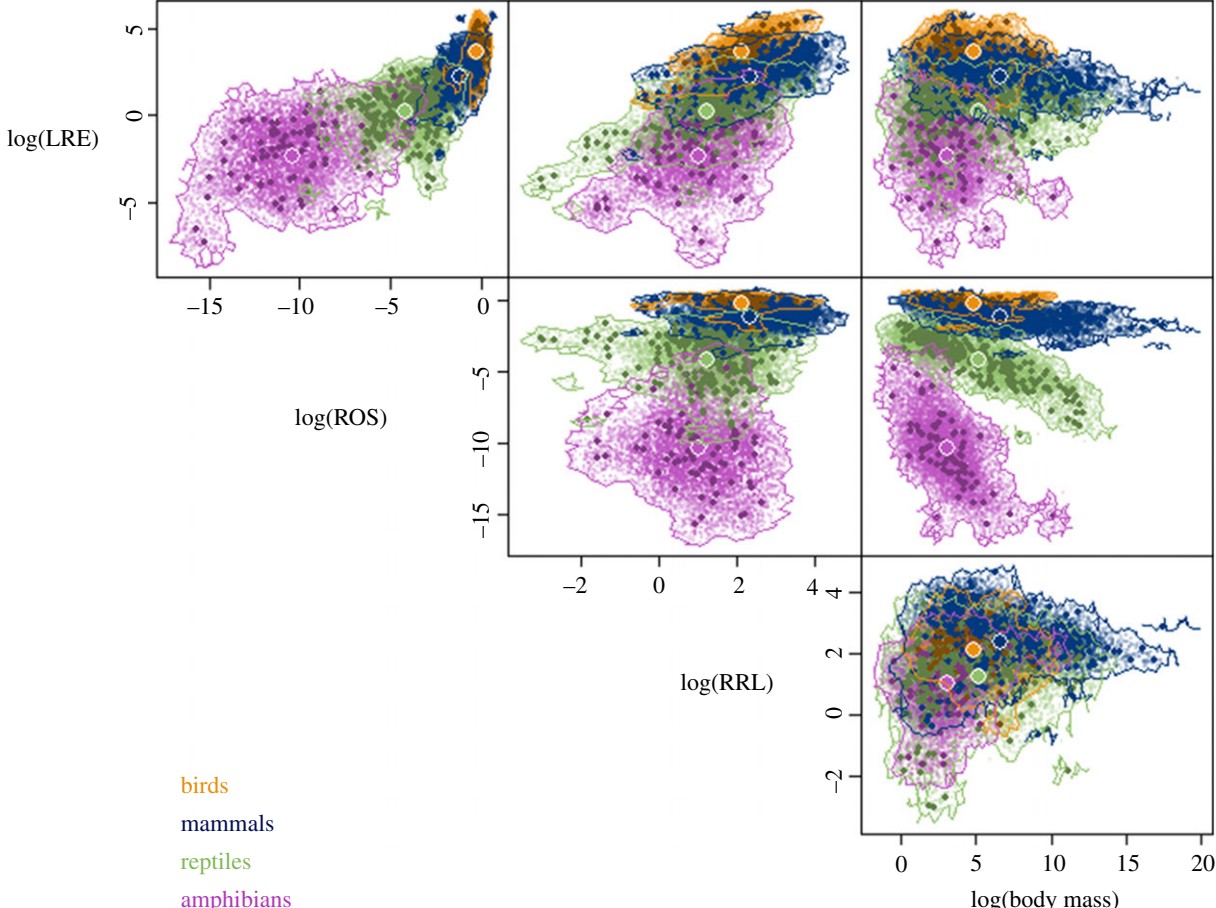

**Figure 3.** Gaussian hypervolumes for the four classes of tetrapods. Large coloured points represent the centroids of each hypervolume. Small dark points represent trait values for individual species while small light points represent random points. The volume of bird hypervolume is 36.69, mammal is 214.98, reptile is 657.93 and amphibian is 1034.54. (Online version in colour.)

**Table 4.** Sorensen similarity of hypervolumes between classes. The Sorensen similarity metric ranges from 0, when hypervolumes are disjunct, to 1 when they are completely overlapping.

|          | Mammalia | Reptilia | Amphibia |
|----------|----------|----------|----------|
| Aves     | 0.14     | 0.0019   | 0.00     |
| Mammalia |          | 0.056    | 0.00     |
| Reptilia |          |          | 0.027    |

adult size (lowest ROS; figure 1d). Many of these differences reflect the constraints imposed by a non-amniote egg. Amniotic eggs contain unique membranes which allow them to be much larger and maintain higher rates of respiration than those of amphibians, which are limited by the rate of diffusion of oxygen through the egg [13,46]. Because offspring can spend longer in the egg, amniotes supply their eggs with substantial yolks that allow offspring to develop to a greater degree before hatching [14]. These adaptations allow amniotes to emerge at a higher stage of development than amphibian offspring do [14] and may help amniotes reach reproductive maturity more quickly—and spend a greater proportion of their total lifespan reproducing—compared to amphibians. These impacts of the amniote egg on the size of the offspring and the extra investment parents provide their offspring in yolk may also explain why amniotes exhibit higher levels of lifetime reproductive investment and ROS than amphibians (figure 1b,d).

While the vast majority of comparative life-history research focuses on amniotes [2,18,47–49], our work highlights the incredible diversity of amphibian life-history strategies compared to other tetrapods. Amphibians occupy a region of the life-history trait space that is almost completely unique from the other three classes. Moreover, the amphibian life-history hypervolume is both the largest in volume—by almost an order of magnitude (figure 3)—and the lowest phylogenetic signal (table 1), indicating that amphibians possess remarkable diversity and evolutionary flexibility in their strategies. This diversity may be driven by the variety of modifications to amphibian life cycles, from neotony to viviparity, which permit variation in clutch size and offspring size [50].

### (b) Endotherms versus ectotherms: the impact of thermic strategy

While the amphibians exhibit the greatest range of life-history metric combinations, both ectothermic classes occupy regions of trait space many orders of magnitude larger than those occupied by the endothermic classes (figure 3). This pattern is consistent with the hypothesis that, while endothermy conveys advantages [16], it also comes with costs that can constrain life-history strategies [48,51,52]. Endothermy is related to higher metabolic power [31], higher potential for production and a greater ability to maintain activity under a broader range of conditions [16,53]. These advantages may allow endotherms to have more resources for both

reproduction [54] and survival by decreasing adult mortality through impacts on foraging durations and predator avoidance [55,56]. Endothermy is energetically expensive [55,57]; however, and it is especially difficult for small organisms to maintain the required thermal differential with their environment. These constraints and advantages have the potential to alter the viability of different life-history strategies via their impact on reproductive allocation and survival. Our results suggest that these advantages and constraints conferred by endothermy have generated lower flexibility in life-history strategies among endothermic species.

The life-history pattern in ROS exhibits the strongest constraint in endotherms. Mammals and birds produce offspring that must attain a greater proportion of their adult mass before independence, which requires greater parental investment. Not only is mean ROS higher for endotherms than ectotherms, the endotherms also demonstrate much less variation, implying that endothermy may constrain the possible range of ROS. The need for greater offspring size for endotherms could reflect the fact that thermogenic tissue is expensive to produce and that offspring may need greater levels of parental investment to produce it [58]. It is also difficult and energetically costly for small individuals to maintain a thermal differential with the environment, which could necessitate greater parental investment to help offspring reach a sufficient size to reduce their thermoregulatory costs [48,54]. Because resources for reproduction are limited, increased investment in offspring necessitates decreases in the number of offspring produced [59]. Thus, this need to invest in larger offspring may preclude endotherms from life-history strategies that produce many small, mostly independent offspring requiring minimal parental care. Without the constraints imposed by endothermy, ectotherms can take on a wider range of life-history traits, including decreased offspring size and increased fecundity [48,51,52].

## (c) Taking flight: the importance of volancy

Volancy drives changes in longevity and parental investment that impose strong constraints on the three dimensionless metrics. Lifespan is longer in birds, as well as in volant mammals, compared to non-volant mammals [17,18]. Flight enables organisms to escape predation more successfully [17,60], which in turn decreases extrinsic mortality and increases longevity. However, a flight is also energetically costly and entails physical stress forces that terrestrial animals do not experience which could impact parental care. The unique skeletal trade-off requiring bones strong enough to endure the higher shearing stress of flight but light enough to reduce flight costs [61,62] could delay independence by requiring offspring to be closer to adult size before becoming independent. In general, volant species must allocate more energy to parental care in order to supply the young with food prior to independence [19]. These effects of flight lead to a variety of changes in the three metrics in both birds and bats, although these two clades also face separate constraints that affect their life-history strategies differently.

Despite the increase in longevity driven by flight, birds have slightly lower RRL values than all mammals, including bats. If birds had similar ages at female maturity as mammals, we would expect RRL to be higher in birds due to this longer lifespan. We observed the opposite, however, with mammals having a significantly higher mean RRL value, indicating that birds take relatively longer to mature despite having longer overall lifespans. Volancy itself does not appear to cause lower RRL, however, since bats have RRL values more similar to mammals than to birds. Instead, it appears as though birds have unique constraints on breeding not faced by mammals. Almost all small birds wait at least 1 year from hatching before they begin breeding, while small mammals do not. Several factors may play a role in this discrepancy. First, migration is more common in small birds than in small mammals, and this additional life-history stage constrains the timing of breeding [63]. Second, even in non-migratory species, birds have a higher metabolic rate and body temperature than mammals [64]. Since reproduction typically necessitates an additional increase in metabolic rate [19], birds have more seasonal constraints on breeding due to the need to maintain the incredibly high energy investment in breeding. Finally, since birds have lower mortality rates in general, they may be able to afford a longer period of investment in growth before beginning reproduction, as predicted by Charnov's evolutionary model [7]. This collection of factors may contribute to lower avian RRL values compared to mammals.

Birds display the highest LRE values of all four tetrapod classes. Volancy is the most important driver of increased lifespan in endotherms, which leads to increases in LRE by increasing the amount of time birds can devote to reproduction over their lifetime [18]. Furthermore, birds have the highest metabolic rates and body temperatures of all tetrapods [64]. Reproduction necessitates a further metabolic increase beyond this already high investment [19]. The elevated LRE of birds may reflect that extra cost of avian reproduction compared to that of mammals and ectotherms.

A flight also appears to be a strong constraint on mean ROS values. Bats occupy a range of ROS values much more similar to that of birds than other mammals (electronic supplementary material, figure S3). This shift in bat ROS to be more bird-like may reflect aerodynamic and biomechanical effects of flight that not only require greater parental care, but also sets limits on large body size, generating similar body size distributions for bats and birds [65]. In terms of the trait space defined by the three dimensionless metrics, they resemble birds far more than mammals on the ROS axis, while behaving much more like mammals in their range of LRE values (electronic supplementary material, figure S4). This evidence suggests that flight operates particularly on ROS out of the three metrics: volant organisms must approach adult body mass before they can be independent from their parents.

## 5. Conclusion

Using Charnov's dimensionless life-history metrics, we took a broad-scale macroecological approach to examining the patterns and general constraints driving life-history allocation across four major tetrapod groups. The four tetrapod classes differ drastically in their combinations of LRE, RRL and ROS, indicating that they adopted different strategies to address fundamental life-history trade-offs. Our analyses suggest broad impacts of evolutionary innovations on the life histories of tetrapods, altering the range of life-history solutions available to each group as innovations created new opportunities and new constraints to navigate. Our

results also demonstrate that the Charnov dimensionless life-history metrics are useful tools for exploring the causes and consequences of life-history patterns across clades.

The strength of macroecology is its cross-species approach which allows it to define the expanse of an evolutionary or ecological canvas by using large numbers of individuals or species to define the constraints, limits and central tendencies. Our results suggest that the different clades have different sized life-history canvases to fill and different constraints on how they can combine traits to fill it. Understanding why specific species exhibit certain dimensionless values on that canvas will probably require a more refined approach, including information about complex trade-offs relating to environmental conditions, species interactions, density dependence and environmental fluctuations [66–70]. These types of questions are better suited to a different branch of life-history theory—demographic life-history theory—which uses demographic approaches to understand variation in life history between species or populations over space and time [71–73]. While macroecological and demographic life-history theory has coexisted thus far primarily by ignoring each other, working to combine these approaches may provide unique insights and opportunities to understand how the evolution of new traits changes the constraint space for a clade and drives new opportunities for variation among species in how they navigate the essential functions of survival and reproduction.

Data accessibility. R code for all figures and analyses, as well as data and links to data files, can be found at https://github.com/Kerkhoff Lab/TetrapodLifeHistory and are available from the Dryad Digital Repository: https://doi.org/10.5061/dryad.kd51c5b3q [74].

Authors' Contributions. C.B.M. carried out all data analyses, participated in the design of the study and drafted the manuscript; A.J.K. supported on study design and analyses and critically revised the manuscript; S.K.M.E. supported on study design and critically revised the manuscript. All authors gave final approval for publication and agree to be held accountable for the work performed therein.

Competing interests. We declare we have no competing interests.

Funding. This research was funded by National Science Foundation (NSF) award DEB-1556651 and the 2017 Kenyon Summer Science Scholarship.

Acknowledgements. We are grateful to Dr Natalie Wright for advice on interpreting PGLS, as well as comments on the manuscript. Many thanks to Dr Benjamin Blonder for assistance with the hypervolume figure. Dr Elizabeth Schultz provided very helpful feedback on avian life-history trends specifically. Finally, thank you to Dr Benjamin Blonder, Dr Brian Maitner, and Dr Brian Enquist for feedback on the early stages of the project and to Dr Eric Charnov for encouragement in exploring dimensionless life histories.

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
