## [Peer Review File · Proceedings of the Royal Society B: Biological Sciences]

Review History

RSPB-2020-1534.R0 (Original submission)

Review form: Reviewer 1

Recommendation

Major revision is needed (please make suggestions in comments)

Scientific importance: Is the manuscript an original and important contribution to its field?

Excellent

General interest: Is the paper of sufficient general interest?

Excellent

Quality of the paper: Is the overall quality of the paper suitable?

Marginal

Is the length of the paper justified?

Yes

Should the paper be seen by a specialist statistical reviewer?

No

Do you have any concerns about statistical analyses in this paper? If so, please specify them explicitly in your report.

Yes

It is a condition of publication that authors make their supporting data, code and materials available - either as supplementary material or hosted in an external repository. Please rate, if applicable, the supporting data on the following criteria.

Is it accessible?

No

Is it clear?

No

Is it adequate?

No

Do you have any ethical concerns with this paper?

No

Comments to the Author

In this manuscript the Babich Morrow et al. investigate the evolution of three dimensionless life-history metrics. I found the topic highly interesting and the language in the paper was very good and easy to read. The discussion on linking the differences in life history to the evolution of amniotic egg, endothermy, and flight was particularly interesting. Hence, I think this has the potential to be a very good contribution to Proceedings B. However, I do have some comments, questions and methodological concerns that needs to be addressed and answered before I can recommend the manuscript for publication.

In the construction of the metrics the maximum adult body mass was used and the maximum longevity was used rather than the average, but there was no discussion of how this biased the results.

The relationship between body mass and the life-history metrics is very interesting. However, the method used (the phylogenetic least-squares analysis) is inadequate to my opinion. A fundamental problem with this analysis is that body size appears both in the response and the explanatory variable of the regression. Therefore, measurement error in body size would create a spurious relationship. A regression analysis taking this into account would be quite advanced and I don't think there is available off the shell tools to do this, so it would require to build a model using e.g. Jags or Template Model Builder or equivalent tools. In addition, this would require to have estimates of measurement error in the traits, which I suspect will be impossible to obtain for many of the species. A sensible analysis with the current data and no measurement error is a multivariate phylogenetic mixed model (see Lynch 1991 Evolution, Hadfield and Nakagawa 2009 J. Evol. Biol.). Here one could fit a model with a Brownian motion part and a white noise part (the residuals). The measurement error and the error correlation would feed into the white noise part of the model, and one could estimate the correlated evolution between body size and the life-history metric from the diffusion matrix of the Brownian motion part of the model. Interpreting this model would be straight forward if the phylogenetic signals/heritabilities of the traits are reasonably high, as one could assume that the white noise part is mainly due to measurement error.

I am not familiar with the hypervolume analysis and the Sorensen similarity index, and hence the numbers given in Table 2 are completely meaningless for me without any further explanation. Is it possible to give a brief explanation of what a change of say 0.01 in the Sorensen similarity index would represent. Alternatively, one could maybe give the fraction of unique and overlapping

volume, which seems much more easy to interpret?

Results on “Metric values across classes”: The ANOVA and Tuckey HSD testes do not give much valuable information to my opinion. I would rather have liked to know the difference between the different classes (such as is given by the 98.1% increase in ROS from amphibians to birds) preferably with uncertainty after controlling for phylogeny.

Results on “Correlated evolution”: First, see my above comments how to estimate correlated evolution. Second, I am not convinced that the AIC comparison of the different evolutionary models are particularly interesting. The differences in model fit among taxa are not discussed at all. Instead it would be good if some measure of phylogenetic signal was calculated and discussed (preferably the same measure for all the metrics). There are several such measures to choose from, I prefer the phylogenetic heritability as I find it easy to interpret. The Table 1 does not have an intercept and no measures of uncertainty in the parameters and no explanation of units.

Results on “Hypervolumes”: Part of the observed variation in the traits is due to measurement error. If the amount of measurement error differs among the taxa, this could potentially lead to spurious differences. Because measurement error usually is additive, and the traits are log transformed this could lead to a larger relative measurement error for species with small trait values. Hence, the RLE and ROS of amphibians may be more inflated by measurement error than mammals and birds. It is hard to judge how much of the observed variation is due to measurement error without any measurements of measurement error. Maybe such measurements exists for some of the species, so that its effect on the results can be investigated? Alternatively one could do simulations to understand how large this problem would be with reasonable levels of measurement errors. At least this problem needs to be acknowledge in the manuscript.

Discussion on “Amphibians to amniotes”: What about evolution of parental care to escape the constraint of the amniote on the size at independence?

Discussion on flight: Since much of the discussion is on flight, it might have been good to show the results on bats in the main manuscript rather than in the supplement.

Discussion on the “Relationship between lifetime reproductive effort and relative offspring size”: One of the main arguments of Charnov’s life-history framework is that in non-growing populations the average reproductive output, R_0 , equals 1. Hence, I do not understand the authors when they say the endotherm classes may have higher R_0 than the ectotherm classes. This would mean that either the endotherm classes would be forever growing in number eventually overflowing the earth or that the ectotherm classes would soon go extinct. According to Charnov, a reasonable assumption is that, for most organisms, the R_0 have fluctuated around 1 during most of their evolutionary history. Hence, the difference between endotherms and ectotherms is most probably in their values of S (the probability of surviving to adulthood). It would be very interesting to calculate the values of S predicted by the data according to equation 8 in Charnov (2002, *Evolutionary Ecology Research*). Regarding the correlation between LRE and ROS it must be noted that measurement error in offspring mass and adult mass will create a spurious correlation between the two metrics, as they both includes these measurements. Again it is hard to judge how much this error contributes to the observed correlation without any notion of how large the error is (again one could have used the diffusion matrix estimated using a multivariate mixed model to get a more sound measurement of the correlation, given that the phylogenetic heritability is reasonably high for the traits).

Figure 1 and 3: It would ease interpretation if the numbers on the axis had the non-logged values (though the distance between the tick marks should be kept on log). What is the units of body mass?

Data: Some related species had exactly the same value of the life-history metrics. Why is this? Does it reflect pseudo replication?

Phylogeny: What was the unit of the branch lengths?

Review form: Reviewer 2

Recommendation

Major revision is needed (please make suggestions in comments)

Scientific importance: Is the manuscript an original and important contribution to its field?

Good

General interest: Is the paper of sufficient general interest?

Excellent

Quality of the paper: Is the overall quality of the paper suitable?

Good

Is the length of the paper justified?

Yes

Should the paper be seen by a specialist statistical reviewer?

No

Do you have any concerns about statistical analyses in this paper? If so, please specify them explicitly in your report.

No

It is a condition of publication that authors make their supporting data, code and materials available - either as supplementary material or hosted in an external repository. Please rate, if applicable, the supporting data on the following criteria.

Is it accessible?

Yes

Is it clear?

Yes

Is it adequate?

Yes

Do you have any ethical concerns with this paper?

No

Comments to the Author

This is a very well-presented MS on a currently popular topic, and with potentially interesting results. I do not have too many comments, because it is so very well presented throughout, especially the graphs/tables conveying what would otherwise be rather complex statistical results, all of the most relevant literature is cited.

I'm unsure of the importance of my major points and whether they necessarily threaten the value of some of the conclusions here, because perhaps these issues just need some better clarification

with more robust justifications and citations in text.

The first concerns the justification and history of the use of these particular life-history measures from Charnov, because they do not appear to have been widely adopted in the life-history literature. Most researchers instead continue to use more straightforward dimensional measures, perhaps because they link directly to speed of reproduction and generation time and hence pace-of-life, as well as to fitness and population-level change for the purposes of understanding the eco-evolutionary role of life-histories. I'm also not sure it is necessarily true that all of Charnov's measures are 'dimensionless' in the way that latent variables from multivariate covariance models are (e.g. PCA axes). LRE perhaps is, but RRL is the time to maturity or first reproduction measured in average species lifetime units, and ROS is the mass of offspring in average parental mass units. Or have I misunderstood these metrics?

Secondly, the use of many of the same measured variables in the different Charnov life-history metrics means that it is unsurprising to see many of the correlations presented here. Adult mass is part of the numerator in both LRE and ROS, and thus maybe it is unsurprising to see a negative correlation between these two measures and mass (Table 1)? LRE and ROS might thus be expected to show a 'striking positive correlation (Fig.3)', because of the obvious mass differences between species? As you say, in L.335-340: "ROS is a component of LRE, which calculates reproductive effort as the product of ROS, longevity, and the number of offspring per year. Thus the slope of the correlation between LRE and ROS is approximately the average number of offspring per year across an organism's reproductive lifespan." Therefore, the 'hypervolumes' that distinguish these different species and tetrapod classes are not necessarily composed of independent axes or dimensions, and much of the variation in multiple dimensions could be driven by variation in only one measured life-history trait.

So, bringing together these first two major points, whilst these analyses of Charnov's metrics appear informative regarding life-history shifts following the evolution of the amnion, endotherms and flight, I'm left wondering how all of this links to pace-of-life and eco-evolutionary dynamics (e.g. strength of density dependence), and thus whether much of what is revealed here may in fact be due to differences in relatively few covarying fast vs slow life-history traits.

More specifically, the use of maximum (rather than mean) body mass for amphibia (L.102-3) perhaps needs some more justification. For example, it would be good to see information on symmetrical (normal?) data distributions for body mass in examples of these types of species, and that these distributions hopefully do not vary too much in shape across species. Both of these conditions seem unlikely, and so it is possible that this might affect any comparisons between amphibia using maximum body mass versus other taxa where mean body mass was used. As noted above, LRE and ROS will be systematically smaller in amphibia as a result, won't they? It would therefore be good to understand how much this methodical assumption could have affected the results presented.

Likewise, the use of maximum longevity (rather than mean) for all species (L.104-5) perhaps needs justifying better. Again, information on strength of the covariances between mean versus maximum longevity in these different types of species, and/or whether these covariances systematically varying across these specific types of species, would help convince the reader that the mean and maximum values covary sufficiently that this substitution is acceptable. For example, with RRL (L.107) the older maximum longevity rather than mean age is divided through by age at female maturity, and so this must affect some results here.

L.321-7 - Along with ROS, you could make it clear that body mass in bats is more bird-like than mammal-like, and it is this that also creates many of the bat-bird similarities in Fig.S2, which is actually quite a cool figure. This would seem to be a point worth emphasising, especially given the importance of adult body mass in many of these metrics (see above).

L.289 – references not numbered 14, 15...

L.352 – should perhaps be 'support for'.

Decision letter (RSPB-2020-1534.R0)

01-Sep-2020

Dear Ms Babich Morrow:

I am writing to inform you that your manuscript RSPB-2020-1534 entitled "Macroevolution of dimensionless life history metrics in tetrapods" has, in its current form, been rejected for publication in Proceedings B.

This action has been taken on the advice of referees, who have recommended that substantial revisions are necessary. With this in mind we would be happy to consider a resubmission, provided the comments of the referees are fully addressed. However please note that this is not a provisional acceptance.

Sincerely,
Dr Daniel Costa
mailto: proceedingsb@royalsociety.org

Associate Editor
Comments to Author:

Both reviewers and I like this manuscript, and think it has a lot of potential. However, the reviewers have given detailed and insightful comments on aspects of the methods that need to be clarified. For example, both reviewers are concerned about using maximum rather than mean values. Reviewer 1 in particular has specific suggestions to improve the statistical treatment of the data. I agree with the reviewers that this manuscript is well written and topical.

Reviewer(s)' Comments to Author:

Referee: 1

Comments to the Author(s)

In this manuscript the Babich Morrow et al. investigate the evolution of three dimensionless life-history metrics. I found the topic highly interesting and the language in the paper was very good and easy to read. The discussion on linking the differences in life history to the evolution of amniotic egg, endothermy, and flight was particularly interesting. Hence, I think this has the potential to be a very good contribution to Proceedings B. However, I do have some comments, questions and methodological concerns that needs to be addressed and answered before I can recommend the manuscript for publication.

In the construction of the metrics the maximum adult body mass was used and the maximum longevity was used rather than the average, but there was no discussion of how this biased the results.

The relationship between body mass and the life-history metrics is very interesting. However, the method used (the phylogenetic least-squares analysis) is inadequate to my opinion. A fundamental problem with this analysis is that body size appears both in the response and the explanatory variable of the regression. Therefore, measurement error in body size would create a spurious relationship. A regression analysis taking this into account would be quite advanced and I don't think there is available off the shell tools to do this, so it would require to build a model using e.g. Jags or Template Model Builder or equivalent tools. In addition, this would require to have estimates of measurement error in the traits, which I suspect will be impossible to obtain for many of the species. A sensible analysis with the current data and no measurement error is a multivariate phylogenetic mixed model (see Lynch 1991 *Evolution*, Hadfield and Nakagawa 2009 *J. Evol. Biol.*). Here one could fit a model with a Brownian motion part and a white noise part (the residuals). The measurement error and the error correlation would feed into the white noise part of the model, and one could estimate the correlated evolution between body size and the life-history metric from the diffusion matrix of the Brownian motion part of the model. Interpreting this model would be straight forward if the phylogenetic signals/heritabilities of the traits are reasonably high, as one could assume that the white noise part is mainly due to measurement error.

I am not familiar with the hypervolume analysis and the Sorensen similarity index, and hence the numbers given in Table 2 are completely meaningless for me without any further explanation. Is it possible to give a brief explanation of what a change of say 0.01 in the Sorensen similarity index would represent. Alternatively, one could maybe give the fraction of unique and overlapping volume, which seems much more easy to interpret?

Results on "Metric values across classes": The ANOVA and Tuckey HSD testes do not give much valuable information to my opinion. I would rather have liked to know the difference between the different classes (such as is given by the 98.1% increase in ROS from amphibians to birds) preferably with uncertainty after controlling for phylogeny.

Results on "Correlated evolution": First, see my above comments how to estimate correlated evolution. Second, I am not convinced that the AIC comparison of the different evolutionary models are particularly interesting. The differences in model fit among taxa are not discussed at all. Instead it would be good if some measure of phylogenetic signal was calculated and discussed (preferably the same measure for all the metrics). There are several such measures to choose from, I prefer the phylogenetic heritability as I find it easy to interpret. The Table 1 does not have an intercept and no measurers of uncertainty in the parameters and no explanation of units.

Results on “Hypervolumes”: Part of the observed variation in the traits is due to measurement error. If the amount of measurement error differs among the taxa, this could potentially lead to spurious differences. Because measurement error usually is additive, and the traits are log transformed this could lead to a larger relative measurement error for species with small trait values. Hence, the RLE and ROS of amphibians may be more inflated by measurement error than mammals and birds. It is hard to judge how much of the observed variation is due to measurement error without any measurements of measurement error. Maybe such measurements exists for some of the species, so that its effect on the results can be investigated? Alternatively one could do simulations to understand how large this problem would be with reasonable levels of measurement errors. At least this problem needs to be acknowledge in the manuscript.

Discussion on “Amphibians to amniotes”: What about evolution of parental care to escape the constraint of the amniote on the size at independence?

Discussion on flight: Since much of the discussion is on flight, it might have been good to show the results on bats in the main manuscript rather than in the supplement.

Discussion on the “Relationship between lifetime reproductive effort and relative offspring size”: One of the main arguments of Charnov’s life-history framework is that in non-growing populations the average reproductive output, R_0 , equals 1. Hence, I do not understand the authors when they say the endotherm classes may have higher R_0 than the ectotherm classes. This would mean that either the endotherm classes would be forever growing in number eventually overflowing the earth or that the ectotherm classes would soon go extinct. According to Charnov, a reasonable assumption is that, for most organisms, the R_0 have fluctuated around 1 during most of their evolutionary history. Hence, the difference between endotherms and ectotherms is most probably in their values of S (the probability of surviving to adulthood). It would be very interesting to calculate the values of S predicted by the data according to equation 8 in Charnov (2002, *Evolutionary Ecology Research*). Regarding the correlation between LRE and ROS it must be noted that measurement error in offspring mass and adult mass will create a spurious correlation between the two metrics, as they both includes these measurements. Again it is hard to judge how much this error contributes to the observed correlation without any notion of how large the error is (again one could have used the diffusion matrix estimated using a multivariate mixed model to get a more sound measurement of the correlation, given that the phylogenetic heritability is reasonably high for the traits).

Figure 1 and 3: It would ease interpretation if the numbers on the axis had the non-logged values (though the distance between the tick marks should be kept on log). What is the units of body mass?

Data: Some related species had exactly the same value of the life-history metrics. Why is this? Does it reflect pseudo replication?

Phylogeny: What was the unit of the branch lengths?

Referee: 2

Comments to the Author(s)

This is a very well-presented MS on a currently popular topic, and with potentially interesting results. I do not have to many comments, because it is so very well presented throughout, especially the graphs/ tables conveying what would otherwise be rather complex statistical results, all of the most relevant literature is cited.

I’m unsure of the importance of my major points and whether they necessarily threaten the value of some of the conclusions here, because perhaps these issues just need some better clarification with more robust justifications and citations in text.

The first concerns the justification and history of the use of these particular life-history measures from Charnov, because they do not appear to have been widely adopted in the life-history literature. Most researchers instead continue to use more straightforward dimensional measures, perhaps because they link directly to speed of reproduction and generation time and hence pace-of-life, as well as to fitness and population-level change for the purposes of understanding the eco-evolutionary role of life-histories. I'm also not sure it is necessarily true that all of Charnov's measures are 'dimensionless' in the way that latent variables from multivariate covariance models are (e.g. PCA axes). LRE perhaps is, but RRL is the time to maturity or first reproduction measured in average species lifetime units, and ROS is the mass of offspring in average parental mass units. Or have I misunderstood these metrics?

Secondly, the use of many of the same measured variables in the different Charnov life-history metrics means that it is unsurprising to see many of the correlations presented here. Adult mass is part of the numerator in both LRE and ROS, and thus maybe it is unsurprising to see a negative correlation between these two measures and mass (Table 1)? LRE and ROS might thus be expected to show a 'striking positive correlation (Fig.3)', because of the obvious mass differences between species? As you say, in L.335-340: "ROS is a component of LRE, which calculates reproductive effort as the product of ROS, longevity, and the number of offspring per year. Thus the slope of the correlation between LRE and ROS is approximately the average number of offspring per year across an organism's reproductive lifespan." Therefore, the 'hypervolumes' that distinguish these different species and tetrapod classes are not necessarily composed of independent axes or dimensions, and much of the variation in multiple dimensions could be driven by variation in only one measured life-history trait.

So, bringing together these first two major points, whilst these analyses of Charnov's metrics appear informative regarding life-history shifts following the evolution of the amnion, endotherms and flight, I'm left wondering how all of this links to pace-of-life and eco-evolutionary dynamics (e.g. strength of density dependence), and thus whether much of what is revealed here may in fact be due to differences in relatively few covarying fast vs slow life-history traits.

More specifically, the use of maximum (rather than mean) body mass for amphibia (L.102-3) perhaps needs some more justification. For example, it would be good to see information on symmetrical (normal?) data distributions for body mass in examples of these types of species, and that these distributions hopefully do not vary too much in shape across species. Both of these conditions seem unlikely, and so it is possible that this might affect any comparisons between amphibia using maximum body mass versus other taxa where mean body mass was used. As noted above, LRE and ROS will be systematically smaller in amphibia as a result, won't they? It would therefore be good to understand how much this methodical assumption could have affected the results presented.

Likewise, the use of maximum longevity (rather than mean) for all species (L.104-5) perhaps needs justifying better. Again, information on strength of the covariances between mean versus maximum longevity in these different types of species, and/or whether these covariances systematically varying across these specific types of species, would help convince the reader that the mean and maximum values covary sufficiently that this substitution is acceptable. For example, with RRL (L.107) the older maximum longevity rather than mean age is divided through by age at female maturity, and so this must affect some results here.

L.321-7 – Along with ROS, you could make it clear that body mass in bats is more bird-like than mammal-like, and it is this that also creates many of the bat-bird similarities in Fig.S2, which is actually quite a cool figure. This would seem to be a point worth emphasising, especially given the importance of adult body mass in many of these metrics (see above).

L.289 – references not numbered 14, 15...

L.352 – should perhaps be ‘support for’.

Author's Response to Decision Letter for (RSPB-2020-1534.R0)

See Appendix A.

RSPB-2021-0200.R0

Review form: Reviewer 3

Recommendation

Major revision is needed (please make suggestions in comments)

Scientific importance: Is the manuscript an original and important contribution to its field?

Excellent

General interest: Is the paper of sufficient general interest?

Excellent

Quality of the paper: Is the overall quality of the paper suitable?

Good

Is the length of the paper justified?

Yes

Should the paper be seen by a specialist statistical reviewer?

Yes

Do you have any concerns about statistical analyses in this paper? If so, please specify them explicitly in your report.

Yes

It is a condition of publication that authors make their supporting data, code and materials available - either as supplementary material or hosted in an external repository. Please rate, if applicable, the supporting data on the following criteria.

Is it accessible?

Yes

Is it clear?

Yes

Is it adequate?

Yes

Do you have any ethical concerns with this paper?

No

Comments to the Author

This study describes life history variation across terrestrial vertebrate classes, using the classic framework proposed by Eric Charnov in a series of papers. I enjoyed reading the MS, as I enjoyed reading previous work by Charnov, and I think the authors have made a brave attempt to bring back the debate regarding the relevance of using invariant metrics in life history analyses. I find this useful. However, I concur with the reviewers that they fall short in convincing that Charnov framework significantly enriches current theory. In the new version, the authors successfully address some of the issues raised by the reviewers, but I feel others still need additional work:

Theoretical framework: The framework proposed by Charnov is used without much critical appraisal, even though previous suggestions that the existence of invariants could be an illusion of regressing life history traits against themselves (e.g. Nee et al. 2005 Science). Referee #2 also makes a valid point when arguing that Charnov metrics have not been widely accepted because they have not so obvious demographic implications. Although I agree that the fact that other traits are not so popular does not mean that they are invalid, I think that the ecological validity still needs to be verified. This would mean that the invariants are not just illusions but relevant ways to describe biological systems. I also agree with the reviewer that RRL is not dimensionless; although the units are different, they still represent time periods. This is relevant given the emphasis of the study in the need to focus on invariants. Finally, there is a notorious lack of integration of the results with well-established mechanisms of life history evolution, described for instance in the work by David Reznik, Stephen Stearns and others.

Methodological issues: An issue raised by the two reviewers, and with which I concur, is that the correlation between the different life history metrics is unsurprising given that these are often derived from the same measured variables. The fact that this issue is only relevant for some comparisons but not others does not exclude the problem. Reviewer #1 suggest to use multiresponse MCMCglmm to avoid creating a spurious relationship between body mass and life-history metrics. I'm not sure whether the mvMORPH approach used by the authors deals with the problem, as this is still uses GLS to fit linear models where the errors are allowed to be phylogenetically correlated. The authors also include estimations of measurement error in the models, yet it is unclear to me how this was done. I looked for the code in their scripts but I could not find it probably because the authors have not yet updated the files at GitHub. In general, I think the analytical approaches used should be better explained (functions used, variables used as response and predictors) and justified (For example, the authors fit a variety of evolutionary models to the data, yet why they do not explain why they need so and what are the evolutionary implications).

Interpretation of results: The evolutionary interpretations of why the studied lineages vary in life history are post hoc explanations difficult to support empirically, given that evidence comes from only one or two evolutionary transitions. These limitations should be clearly discussed in the text.

Specific issues:

L221: What data? How did you estimate measurement error?

L223: Was body size included as predictor or response?

Decision letter (RSPB-2021-0200.R0)

02-Mar-2021

Dear Ms Babich Morrow:

Your manuscript has now been peer reviewed and the reviews have been assessed by an Associate Editor. The reviewers' comments (not including confidential comments to the Editor) and the comments from the Associate Editor are included at the end of this email for your reference. As you will see, the reviewers and the Editors have raised some concerns with your manuscript and we would like to invite you to revise your manuscript to address them.

Research ethics:

Use of animals and field studies:

It is a condition of publication that you make available the data and research materials supporting the results in the article (<https://royalsociety.org/journals/authors/author-guidelines/#data>). Datasets should be deposited in an appropriate publicly available repository and details of the associated accession number, link or DOI to the datasets must be included in the Data Accessibility section of the article (<https://royalsociety.org/journals/ethics-policies/data-sharing-mining/>). Reference(s) to datasets should also be included in the reference list of the article with DOIs (where available).

Please submit a copy of your revised paper within three weeks. If we do not hear from you within this time your manuscript will be rejected. If you are unable to meet this deadline please let us know as soon as possible, as we may be able to grant a short extension.

Best wishes,
Dr Daniel Costa
mailto:proceedingsb@royalsociety.org

Associate Editor Board Member

Comments to Author:

Thank you for addressing many reviewer comments and explaining the changes carefully. This manuscript is improved. You have dealt with many of the issues, but there are still some important problems that have not been adequately addressed and would need to be before the manuscript could be accepted- the most important for me are the methodological issues of how to treat multicollinearity and clarity about the methods including code. I agree with the reviewer that there are valuable aspects of this analysis, and it is positive for the field to discuss Charnov's metrics in a new light.

Reviewer(s)' Comments to Author:

Referee: 3

Comments to the Author(s).

This study describes life history variation across terrestrial vertebrate classes, using the classic framework proposed by Eric Charnov in a series of papers. I enjoyed reading the MS, as I enjoyed reading previous work by Charnov, and I think the authors have made a brave attempt to bring back the debate regarding the relevance of using invariant metrics in life history analyses. I find this useful. However, I concur with the reviewers that they fall short in convincing that Charnov framework significantly enriches current theory. In the new version, the authors successfully address some of the issues raised by the reviewers, but I feel others still need additional work:

Theoretical framework: The framework proposed by Charnov is used without much critical appraisal, even though previous suggestions that the existence of invariants could be an illusion

of regressing life history traits against themselves (e.g. Nee et al. 2005 Science). Referee #2 also makes a valid point when arguing that Charnov metrics have not been widely accepted because they have not so obvious demographic implications. Although I agree that the fact that other traits are not so popular does not mean that they are invalid, I think that the ecological validity still needs to be verified. This would mean that the invariants are not just illusions but relevant ways to describe biological systems. I also agree with the reviewer that RRL is not dimensionless; although the units are different, they still represent time periods. This is relevant given the emphasis of the study in the need to focus on invariants. Finally, there is a notorious lack of integration of the results with well-established mechanisms of life history evolution, described for instance in the work by David Reznik, Stephen Stearns and others.

Methodological issues: An issue raised by the two reviewers, and with which I concur, is that the correlation between the different life history metrics is unsurprising given that these are often derived from the same measured variables. The fact that this issue is only relevant for some comparisons but not others does not exclude the problem. Reviewer #1 suggest to use multiresponse MCMCglmm to avoid creating a spurious relationship between body mass and life-history metrics. I'm not sure whether the mvMORPH approach used by the authors deals with the problem, as this is still uses GLS to fit linear models where the errors are allowed to be phylogenetically correlated. The authors also include estimations of measurement error in the models, yet it is unclear to me how this was done. I looked for the code in their scripts but I could not find it probably because the authors have not yet updated the files at GitHub. In general, I think the analytical approaches used should be better explained (functions used, variables used as response and predictors) and justified (For example, the authors fit a variety of evolutionary models to the data, yet why they do not explain why they need so and what are the evolutionary implications).

Interpretation of results: The evolutionary interpretations of why the studied lineages vary in life history are post hoc explanations difficult to support empirically, given that evidence comes from only one or two evolutionary transitions. These limitations should be clearly discussed in the text.

Specific issues:

L221: What data? How did you estimate measurement error?

L223: Was body size included as predictor or response?

Author's Response to Decision Letter for (RSPB-2021-0200.R0)

See Appendix B.

Decision letter (RSPB-2021-0200.R1)

06-Apr-2021

Dear Ms Babich Morrow

I am pleased to inform you that your manuscript entitled "Macroevolution of dimensionless life history metrics in tetrapods" has been accepted for publication in Proceedings B.

Data Accessibility section

Open Access

Paper charges

Sincerely,

Dr Daniel Costa

Appendix A

Dear Editors,

We are submitting a follow-up revision to our manuscript titled “Macroevolution of dimensionless life history metrics in tetrapods” to be considered for publication in Proceedings of the Royal Society B.

Thank you so much for your constructive feedback on our work and the opportunity for further revision. We appreciated both the methodological and theoretical critiques. To address the methodological concerns, we restructured the paper accordingly to better clarify the desired purposes of our analyses: (1) to examine variation in these life history metrics within each class, and (2) to examine variation between the classes. This framing aimed to also highlight the theoretical justification for exploring this topic, namely that these metrics, regardless of invariance, can be used to compare changes in life history strategies across organisms spanning a wide range in body size. Below, we respond to each reviewer comment in detail. Following the response to comments, we have attached a tracked changes version of the manuscript

Thank you for your consideration of our revision.

Sincerely,

Cecina Babich Morrow, Morgan Ernest, Drew Kerkhoff

REVIEWS

Referee 3:

This study describes life history variation across terrestrial vertebrate classes, using the classic framework proposed by Eric Charnov in a series of papers. I enjoyed reading the MS, as I enjoyed reading previous work by Charnov, and I think the authors have made a brave attempt to bring back the debate regarding the relevance of using invariant metrics in life history analyses. I find this useful. However, I concur with the reviewers that they fall short in convincing that Charnov framework significantly enriches current theory. In the new version, the authors successfully address some of the issues raised by the reviewers, but I feel others still need additional work:

In restructuring the paper, we have highlighted that, regardless of the invariance of the metrics, they vary dramatically between the four tetrapod classes in ways that are consistent with our knowledge of the distinguishing adaptations of each class. Since the metrics are dimensionless, they permit comparison of strategies across groups that span a large range of body sizes. We argue that these strengths make the dimensionless metrics a useful tool with which to compare life history strategies between groups.

Theoretical framework: The framework proposed by Charnov is used without much critical appraisal, even though previous suggestions that the existence of invariants could be an illusion of regressing life history traits against themselves (e.g. Nee et al. 2005 Science). Referee #2 also makes a valid point when arguing that Charnov metrics have not been widely accepted because they have not so obvious demographic implications. Although I agree that the fact that other traits are not so popular does not mean that they are invalid, I think that the ecological validity still needs to be verified. This would mean that the invariants are not just illusions but relevant ways to describe biological systems.

Through our revision of the introduction, we clarify that examining body mass invariance is not the chief goal of this paper. Instead, we are interested in testing Charnov's hypothesis of a life history cube, in which different taxa occupy different regions of life history space. In the conclusion, we have expanded on why a macroecological approach, i.e. examining patterns in these metrics across major tetrapod classes, can complement demographic life history analyses. While these analyses are outside of the scope of this paper, we hope that this can be the starting point of future work that can combine macroecological and demographic life history approaches to investigate how life history trait evolution can affect trade-offs within a clade.

I also agree with the reviewer that RRL is not dimensionless; although the units are different, they still represent time periods. This is relevant given the emphasis of the study in the need to focus on invariants.

RRL is calculated by dividing adult lifespan by time to sexual maturity:

- 1. $RRL = \text{adult lifespan (yr)} / \text{time to sexual maturity (yr)}$*

Both adult lifespan and time to sexual maturity are measured in years or days (or any other unit of time). Thus the units cancel out, rendering the resulting metric dimensionless.

We clarified the methods section for all three metrics to display the equations for each metric. These equations will help the reader clearly visualize why each of the three metrics is dimensionless. Just as ROS is a dimensionless measure of size, RRL is a dimensionless measure of time. Both are a ratio of two quantities measured in the same units, and thus both are dimensionless.

Finally, there is a notorious lack of integration of the results with well-established mechanisms of life history evolution, described for instance in the work by David Reznik, Stephen Stearns and others.

We have reworked the introduction and conclusion to situate this work in relation to other life history research. In this study, we take a macroecological approach to life history research by investigating the large-scale patterns in life history strategy displayed across tetrapods. This approach does not attempt to replace other methods of investigating life history evolution; rather, we hope that we can provide a complementary approach that can both inform and be informed by other areas of life history research.

Methodological issues: An issue raised by the two reviewers, and with which I concur, is that the correlation between the different life history metrics is unsurprising given that these are often derived from the same measured variables. The fact that this issue is only relevant for some comparisons but not others does not exclude the problem. Reviewer #1 suggest to use multiresponse MCMCglmm to avoid creating a spurious relationship between body mass and life-history metrics. I'm not sure whether the mvMORPH approach used by the authors deals with the problem, as this is still uses GLS to fit linear models where the errors are allowed to be phylogenetically correlated. The authors also include estimations of measurement error in the models, yet it is unclear to me how this was done.

The multivariate pglms approach used in the mvMorph package allows the estimation of measurement error as a nuisance parameter, as in mixed models (Housworth et al. 2004; Clavel et al. 2019). This approach was taken at the request of a previous reviewer, and we have tried to clarify the reasoning behind the approach in the revised methods.

The fact that some of the metrics are correlated may be unsurprising, but the more interesting result is that they are not necessarily correlated in a straightforward or artifactual way. For example, LRE and ROS share adult body size in the denominator of their calculation - so if their correlation was simply driven by their shared use of body size, they should be linearly related, with a slope of 1 on a log-log plot. But this is not the case across taxa, as can be seen in the nonlinear variation observed across taxa in figure 3. The complex relationships among the components of the metrics means that they may be correlated in complex ways, especially across taxa that solve life history problems in very different ways. The larger point though, and one that we have tried to highlight more fully in both the introduction and methods, is that while the metrics make use of some common component life history measures, each is meant to represent a particular

sort of life history trade-off and has a meaningful biological interpretation. Thus, we thought it important to explore how they are correlated.

I looked for the code in their scripts but I could not find it probably because the authors have not yet updated the files at GitHub.

We have updated all code on GitHub:

<https://github.com/KerkhoffLab/TetrapodLifeHistory>.

In general, I think the analytical approaches used should be better explained (functions used, variables used as response and predictors) and justified (For example, the authors fit a variety of evolutionary models to the data, yet why they do not explain why they need so and what are the evolutionary implications).

We clarified the reasoning behind our analyses in the methods section to structure our results around the following two groups of questions: (1) how the metrics vary within classes, and (2) how the metrics vary between classes. To examine variation within tetrapod classes, we calculated phylogenetic signal, performed both univariate and multivariate PGLS and analyze the resulting correlation matrix between metrics. To examine variation between classes, we performed ANOVA, simulated trait evolution across the tetrapod phylogeny, and created hypervolumes. This restructuring of the methods and results section aims to address this feedback by clarifying the purposes of the analytical approaches we selected. Additionally, we clarified figure captions and result text to highlight how each variable was used in the models.

Interpretation of results: The evolutionary interpretations of why the studied lineages vary in life history are post hoc explanations difficult to support empirically, given that evidence comes from only one or two evolutionary transitions. These limitations should be clearly discussed in the text.

We added a further caveat to the discussion to highlight that we cannot make any definitive links between a given evolutionary transition and a resulting change in life history metrics. However, it is still beneficial to examine how these metrics vary between classes, and whether the variation aligns with our expectations given what we know about the adaptations that each class possesses.

Specific issues:

L221: What data? How did you estimate measurement error?

See comment above.

L223: Was body size included as predictor or response?

Body size was included as a predictor.

Appendix B

Dear Editors,

We are submitting a follow-up revision to our manuscript titled “Macroevolution of dimensionless life history metrics in tetrapods” to be considered for publication in Proceedings of the Royal Society B.

Thank you so much for your constructive feedback on our work and the opportunity for further revision. We appreciated both the methodological and theoretical critiques. To address the methodological concerns, we restructured the paper accordingly to better clarify the desired purposes of our analyses: (1) to examine variation in these life history metrics within each class, and (2) to examine variation between the classes. This framing aimed to also highlight the theoretical justification for exploring this topic, namely that these metrics, regardless of invariance, can be used to compare changes in life history strategies across organisms spanning a wide range in body size. Below, we respond to each reviewer comment in detail. Following the response to comments, we have attached a tracked changes version of the manuscript

Thank you for your consideration of our revision.

Sincerely,

Cecina Babich Morrow, Morgan Ernest, Drew Kerkhoff

REVIEWS

Referee 3:

This study describes life history variation across terrestrial vertebrate classes, using the classic framework proposed by Eric Charnov in a series of papers. I enjoyed reading the MS, as I enjoyed reading previous work by Charnov, and I think the authors have made a brave attempt to bring back the debate regarding the relevance of using invariant metrics in life history analyses. I find this useful. However, I concur with the reviewers that they fall short in convincing that Charnov framework significantly enriches current theory. In the new version, the authors successfully address some of the issues raised by the reviewers, but I feel others still need additional work:

In restructuring the paper, we have highlighted that, regardless of the invariance of the metrics, they vary dramatically between the four tetrapod classes in ways that are consistent with our knowledge of the distinguishing adaptations of each class. Since the metrics are dimensionless, they permit comparison of strategies across groups that span a large range of body sizes. We argue that these strengths make the dimensionless metrics a useful tool with which to compare life history strategies between groups.

Theoretical framework: The framework proposed by Charnov is used without much critical appraisal, even though previous suggestions that the existence of invariants could be an illusion of regressing life history traits against themselves (e.g. Nee et al. 2005 Science). Referee #2 also makes a valid point when arguing that Charnov metrics have not been widely accepted because they have not so obvious demographic implications. Although I agree that the fact that other traits are not so popular does not mean that they are invalid, I think that the ecological validity still needs to be verified. This would mean that the invariants are not just illusions but relevant ways to describe biological systems.

Through our revision of the introduction, we clarify that examining body mass invariance is not the chief goal of this paper. Instead, we are interested in testing Charnov's hypothesis of a life history cube, in which different taxa occupy different regions of life history space. In the conclusion, we have expanded on why a macroecological approach, i.e. examining patterns in these metrics across major tetrapod classes, can complement demographic life history analyses. While these analyses are outside of the scope of this paper, we hope that this can be the starting point of future work that can combine macroecological and demographic life history approaches to investigate how life history trait evolution can affect trade-offs within a clade.

I also agree with the reviewer that RRL is not dimensionless; although the units are different, they still represent time periods. This is relevant given the emphasis of the study in the need to focus on invariants.

RRL is calculated by dividing adult lifespan by time to sexual maturity:

1. $RRL = \text{adult lifespan (yr)} / \text{time to sexual maturity (yr)}$

Both adult lifespan and time to sexual maturity are measured in years or days (or any other unit of time). Thus the units cancel out, rendering the resulting metric dimensionless.

We clarified the methods section for all three metrics to display the equations for each metric. These equations will help the reader clearly visualize why each of the three metrics is dimensionless. Just as ROS is a dimensionless measure of size, RRL is a dimensionless measure of time. Both are a ratio of two quantities measured in the same units, and thus both are dimensionless.

Finally, there is a notorious lack of integration of the results with well-established mechanisms of life history evolution, described for instance in the work by David Reznik, Stephen Stearns and others.

We have reworked the introduction and conclusion to situate this work in relation to other life history research. In this study, we take a macroecological approach to life history research by investigating the large-scale patterns in life history strategy displayed across tetrapods. This approach does not attempt to replace other methods of investigating life history evolution; rather, we hope that we can provide a complementary approach that can both inform and be informed by other areas of life history research.

Methodological issues: An issue raised by the two reviewers, and with which I concur, is that the correlation between the different life history metrics is unsurprising given that these are often derived from the same measured variables. The fact that this issue is only relevant for some comparisons but not others does not exclude the problem. Reviewer #1 suggest to use multiresponse MCMCglmm to avoid creating a spurious relationship between body mass and life-history metrics. I'm not sure whether the mvMORPH approach used by the authors deals with the problem, as this is still uses GLS to fit linear models where the errors are allowed to be phylogenetically correlated. The authors also include estimations of measurement error in the models, yet it is unclear to me how this was done.

The multivariate pglS approach used in the mvMorph package allows the estimation of measurement error as a nuisance parameter, as in mixed models (Housworth et al. 2004; Clavel et al. 2019). This approach was taken at the request of a previous reviewer, and we have tried to clarify the reasoning behind the approach in the revised methods.

The fact that some of the metrics are correlated may be unsurprising, but the more interesting result is that they are not necessarily correlated in a straightforward or artifactual way. For example, LRE and ROS share adult body size in the denominator of their calculation - so if their correlation was simply driven by their shared use of body size, they should be linearly related, with a slope of 1 on a log-log plot. But this is not the case across taxa, as can be seen in the nonlinear variation observed across taxa in figure 3. The complex relationships among the components of the metrics means that they may be correlated in complex ways, especially across taxa that solve life history problems in very different ways. The larger point though, and one that we have tried to highlight more fully in both the introduction and methods, is that while the metrics make use of some common component life history measures, each is meant to represent a particular

sort of life history trade-off and has a meaningful biological interpretation. Thus, we thought it important to explore how they are correlated.

I looked for the code in their scripts but I could not find it probably because the authors have not yet updated the files at GitHub.

We have updated all code on GitHub:

<https://github.com/KerkhoffLab/TetrapodLifeHistory>.

In general, I think the analytical approaches used should be better explained (functions used, variables used as response and predictors) and justified (For example, the authors fit a variety of evolutionary models to the data, yet why they do not explain why they need so and what are the evolutionary implications).

We clarified the reasoning behind our analyses in the methods section to structure our results around the following two groups of questions: (1) how the metrics vary within classes, and (2) how the metrics vary between classes. To examine variation within tetrapod classes, we calculated phylogenetic signal, performed both univariate and multivariate PGLS and analyze the resulting correlation matrix between metrics. To examine variation between classes, we performed ANOVA, simulated trait evolution across the tetrapod phylogeny, and created hypervolumes. This restructuring of the methods and results section aims to address this feedback by clarifying the purposes of the analytical approaches we selected. Additionally, we clarified figure captions and result text to highlight how each variable was used in the models.

Interpretation of results: The evolutionary interpretations of why the studied lineages vary in life history are post hoc explanations difficult to support empirically, given that evidence comes from only one or two evolutionary transitions. These limitations should be clearly discussed in the text.

We added a further caveat to the discussion to highlight that we cannot make any definitive links between a given evolutionary transition and a resulting change in life history metrics. However, it is still beneficial to examine how these metrics vary between classes, and whether the variation aligns with our expectations given what we know about the adaptations that each class possesses.

Specific issues:

L221: What data? How did you estimate measurement error?

See comment above.

L223: Was body size included as predictor or response?

Body size was included as a predictor.